# The HD-OCT Study May Be Useful in Searching for Markers of Preclinical Stage of Diabetic Retinopathy in Patients with Type 1 Diabetes

**DOI:** 10.3390/diagnostics9030105

**Published:** 2019-08-26

**Authors:** Magdalena Kołodziej, Arleta Waszczykowska, Irmina Korzeniewska-Dyl, Aleksandra Pyziak-Skupien, Konrad Walczak, Dariusz Moczulski, Piotr Jurowski, Wojciech Młynarski, Agnieszka Szadkowska, Agnieszka Zmysłowska

**Affiliations:** 1Department of Practical Obstetrics, Medical University of Lodz, 90-419 Lodz, Poland; 2Department of Ophthalmology and Vision Rehabilitation, Medical University of Lodz, 90-419 Lodz, Poland; 3Department of Internal Medicine and Nephrodiabetology, Medical University of Lodz, 90-419 Lodz, Poland; 4Department of Pediatrics, Silesian Medical University in Katowice, 40-055 Katowice, Poland; 5Department of Pediatrics, Oncology and Hematology, Medical University of Lodz, 90-419 Lodz, Poland; 6Department of Pediatrics, Diabetology, Endocrinology and Nephrology, Medical University of Lodz, 90-419 Lodz, Poland

**Keywords:** diabetic retinopathy, type 1 diabetes, HD-OCT, ONH, RNFL, choroidal thickness

## Abstract

The aim of the study was to analyze the thickness of individual retinal layers in patients with type 1 diabetes (T1D) in comparison to the control group and in relation to markers of diabetes metabolic control. The study group consisted of 111 patients with an average of 6-years of T1D duration. The control group included 36 gender- and age-matched individuals. In all patients optical coherence tomography (OCT) study was performed using HD-OCT Cirrus 5000 with evaluation of optic nerve head (ONH) parameters, thickness of retinal nerve fiber layer (RNFL) with its quadrants, macular full-thickness parameters, ganglion cells with inner plexus layer (GCIPL) and choroidal thickness (CT). Lower disc area value was observed in the study group as compared to controls (*p* = 0.0215). Negative correlations were found both between age at examination and rim area (R = −0.28, *p* = 0.0007) and between superior RNFL thickness and duration of diabetes (R = −0.20, *p* = 0.0336). Positive correlation between center thickness and SD for average glycemia (R = 0.30, *p* = 0.0071) was noted. Temporal CT correlated positively with age at examination (R = 0.21, *p* = 0.0127). The selected parameters the HD-OCT study may in the future serve as potential markers of preclinical phase of DR in patients with T1D.

## 1. Introduction

Type 1 diabetes (T1D) is one of the most common metabolic diseases diagnosed in children and is associated with the risk of development of numerous chronic complications. Diabetic eye disease—as the most common of the chronic complications of T1D—may be connected with abnormalities in various eye structures. However, the complication related to the greatest risk of loss of vision is diabetic retinopathy (DR) and accompanying diabetic macular edema (DME) [1,2]. DR is considered to be the earliest chronic complication of type 1 diabetes because it is present even after several years of its course [3]. The development and progression of diabetic retinopathy is influenced mainly by the degree of metabolic diabetes control, diabetes duration, presence of dyslipidemia, proteinuria, arterial hypertension, pregnancy and genetic predisposition [4]. Many studies have shown that proper control of glycemia, blood pressure and lipid levels may reduce the risk of DR development and significantly slow its progression [5].

Currently, fluorescein angiography (FA) remains the gold standard in the diagnosis of clinical phase of diabetic retinopathy, which enables detailed assessment of blood vessels of the fundus of the eye as an invasive method [6]. FA is a test that involves taking a series of photographs of the fundus after intravenous administration of a contrast agent (fluorescein). FA enables visualization of the circulation in the retinal vessels and indirectly the uvea, as well as evaluation of the state of the retinal pigment epithelium and retinal blood vessels which are not visible during fundus examination. This technique has become fundamental when assessing the complications of diabetic retinopathy, retinal vascular obstruction, or when diagnosing many macular diseases. However, FA is associated with severe potential risks [7].

Thus, in an attempt to optimize diagnostic procedures, the most informative and least invasive diagnostic method should be identified. Optical coherence tomography (OCT) is a safe, non-contact, non-invasive and effective technique of imaging tissue and could be considered as a valid alternative to FA. Intra-retinal leakage is not necessary to confirm the presence of DME, and FA is useful only as a guide for focal or grid laser treatment of thickened areas. Therefore, numerous new diagnostic tools such as high-definition OCT (HD-OCT) and spectral-domain (SD) OCT are being evaluated clinically. These methods provide improved capabilities for the imaging of the intra-retinal morphology and/or allow for the analysis of the structural and functional aspects of leakage activity. Moreover, they offer realistic 3D imaging of retinal and subretinal levels in various diseases [8]. Cirrus HD-OCT (Carl Zeiss Meditec, Inc., Dublin, OH, USA) is an SD-OCT that has an axial resolution of 5 μm and a scan velocity of 27,000 axial scans per second. These features improve the ability to visualize smaller and thinner structures that are difficult to visualize with time-domain OCT (Stratus OCT).

Therefore, alternative imaging methods such as OCT study are suggested in patients, especially considering the new approach to pathologic mechanisms leading to DR development [9]. Until now, diabetic retinopathy has been considered to be a mainly vascular complication, however, it is now known that neurodegenerative processes take place in the retina before the clinical retinopathy, which is its preclinical period [5]. Recent publications have shown that these early symptoms of neuronal damage of the retinal layer are present before the appearance of changes visible at the fundus of the eye and may contribute to the development of microvascular abnormalities [10,11]. These reports also indicate a decrease in retinal thickness in diabetic patients, mainly in macular ganglion cells. This suggests progress in neurodegeneration as an early, preclinical stage of diabetic retinopathy [12,13].

In addition, it is known that previous attempts to treat diabetic retinopathy are being made in its clinical phase [14,15], whereas, from the patient’s point of view, it would be worthwhile to try to stop the progress of retinal neurodegeneration at the earliest possible stage [16]. The lack of unambiguously defined early markers of DR preclinical phase leads to their search based on new and non-invasive imaging studies.

The aim of the study was to evaluate the thickness of retina in its individual layers in patients with type 1 diabetes as compared to the control group and in relation to the parameters of metabolic diabetes control.

## 2. Materials and Methods

The study was conducted in accordance with the Declaration of Helsinki, and the study protocol was approved by the University Bioethics Committee at the Medical University in Lodz, Poland (RNN/374/17/KE, 19/12/2017). Patients and/or their parents gave written informed consent for participation in the study.

The work described has been carried out in accordance with The Declaration of Helsinki of 1975 and revised in 2013 for experiments involving humans.

The study group consisted of 111 patients (F-61/M-50) including 82 children and 29 adults aged 5.6 to 50.8 (min-max) years at the time of study, with an average 6-year course of clinically overt T1D. In all patients, diabetes was diagnosed according to WHO criteria. The presence of autoantibodies characteristic for T1D and decreased C-peptide plasma levels were observed in all patients at T1D onset. Detailed characteristics of the study group are shown in Table 1.

The control group consisted of 36 healthy (F-23/M-13) gender- (*p* = 0.3466) and age-matched (*p* = 0.1688) individuals with no glucose tolerance disturbances.

Patients under 5 years of age with diagnosed arterial hypertension, clinically manifest diabetic retinopathy, diabetic kidney disease and previous ocular surgery, any eye disease, myopia and hyperopia above 3 diopters, chronic use of topical medications were excluded from the study and control groups.

The HD-OCT study was performed in the patients from the study and control groups in the Department of Ophthalmology and Vision Rehabilitation of the Medical University of Lodz, Poland using the Cirrus HD-OCT device (5000: Carl Zeiss Meditec. Inc., Dublin, OH, USA) after mydriasis and evaluated independently by two experienced ophthalmologists.

Two scans, including one macular scan centered on the fovea (macular cube 512 × 128 protocol) and one peripapillary retinal nerve fiber layer (RNFL) scan centered on the optic disc (optic disc cube 200 × 200 protocol) were acquired through dilated pupils. The mean retinal thickness values were obtained on all images for foveal subfield and the inner and outer rings of a standard ETDRS (Early Treatment of Diabetic Retinopathy Study) grid. Macular ganglion cell layer/inner plexiform layer (GCIPL), RNFL (retinal nerve fiber layer) and optic nerve head (ONH) parameters were measured automatically using the internal ganglion cell, RNFL and ONH analysis algorithms, respectively. The following GCIPL thickness measurements were analyzed: Average, minimum and sectoral (superonasal, superior, superotemporal, inferotemporal, inferior and inferonasal). RNFL total thickness as well as superior, inferior, temporal and nasal RNFL thickness were evaluated. For the ONH analysis, the following parameters were included: Disc area, rim area, cup volume and cup-to-disc (c/d) ratio. Choroidal thickness (CT) was measured manually from the outer portion of the hyperreflective line corresponding to the retinal pigment epithelium to the inner surface of the sclera, using the Cirrus linear measurement tool (HD 21-line raster). To be included in this study, images had to be at least 6 of 10 in intensity and taken as close to the fovea as possible, by choosing to image the thinnest point of the macula, with the understanding that slight differences in positioning could affect the measured thickness. Choroidal thickness was measured at the fovea and 1-mm temporal and nasal to the fovea. Averages from measurements in both eyes were calculated and used for further analysis.

The HD-OCT results were related to individual parameters and indicators of clinical course of diabetes including: Age at the time of the study (years), gender, age at diabetes onset (years), duration of diabetes (years), average HbA1c from the last year preceding the study (%), average blood glucose (BG) level from 14 days before an examination (mg/dL), standard deviation (SD) for BG evaluating the fluctuations in BG results from 14 days before an examination, BMI-Z-score and mean insulin uptake (U/kg).

HbA1c was determined by high-performance liquid chromatography (HPLC) using the Bio-Rad VARIANT^TM^ Hemoglobin A1c Program (Bio-Rad Laboratories Inc., Hercules, CA, USA) with its values represented as percentages and in addition as mmol/mol according to IFCC (International Federation of Clinical Chemistry).

### Statistical Analysis

The assessment of the normality of distribution was carried out using the Kolmogorov-Smirnov test and verified by the Shapiro-Wilk test. Comparisons of the HD-OCT variables between study and control groups were performed using a non-parametric Mann-Whitney’s test. For the correlation analysis, a Spearman correlation test was used. Categorical variables were presented as numbers with appropriate percentages and continuous variables as medians with interquartile range (IQR). Results with *p*-values < 0.05 were considered as statistically significant. Analyses were performed using Statistica 13.1 PL software (Statsoft, Tulsa, OK, USA).

## 3. Results

The parameters of optic nerve head (ONH); thickness of retinal nerve fiber layer (RNFL) and its quadrants; average retinal thickness, central retinal thickness, total retinal volume; macular ganglion cell layer/inner plexiform layer (GCIPL) thickness; and choroidal thickness (CT) were compared between the study and control groups. A significantly lower disc area value was observed in the study group (Me 1.84 mm^2^; IQR 1.62–2.04) in comparison to the control group (Me 1.98 mm^2^; IQR 1.82–2.14, *p* = 0.0215) (Table 2). There were no statistically significant differences in other HD–OCT parameters between the studied groups (Table 2).

Then, in the study group the correlations between HD-OCT and individual parameters and markers of clinical course of diabetes were analyzed.

A negative correlation between rim area and age at the time of examination was observed (R = −0.28, *p* = 0.0007) (Figure 1). Moreover, the c/d ratio correlated positively with age of patients at the study time (R = 0.17, *p* = 0.0449). There was also a tendency to negative correlations between rim area and age at T1D onset (R = −0.19, *p* = 0.0517) and disc area and HbA1c value (R = −0.17, *p* = 0.0843).

In the study group, negative correlations between superior RNFL thickness and duration of diabetes (R = −0.20, *p* = 0.0336) (Figure 2) and between inferior RNFL thickness and HbA1c (R = −0.20, *p* = 0.0364) were also noted. There were positive correlations between macular center thickness and SD for average blood glucose level (R = 0.30, *p* = 0.0071).

The age of patients at the study time showed a trend to negative correlations with both average GCIPL thickness (R = −0.15, *p* = 0.0706) and GCIPL lower thickness (R = −0.15, *p* = 0.0759). A trend with respect to a negative correlation between GCIPL low-temporal thickness and HbA1c (R = −0.18, *p* = 0.0696) was also found.

It was also observed that temporal CT correlated positively both with age at the examination time (R = 0.21, *p* = 0.0127) (Figure 3) and age at diabetes diagnosis (R = 0.22, *p* = 0.0255). A positive correlation was also found between the subfoveal CT and the age at diabetes onset (R = 0.23, *p* = 0.0189), while a tendency to a positive correlation between the subfoveal CT and the age at examination (R = 0.16, *p* = 0.0596) was observed. Other parameters of clinical course of type 1 diabetes did not correlate with the HD-OCT parameters (*p* > 0.1).

## 4. Discussion

In the present study, for the first time such a large number of patients with type 1 diabetes, both children and young adults, were examined using the HD-OCT study.

A significant reduction of the disc area in the study group was observed in comparison to healthy controls. Moreover, in patients with T1D the selected ONH parameters correlated with their age. In previous studies, no differences in rim area, disc area or cup to disc ratio between pediatric patients with T1D and control group [17] were found. Only significant differences in c/d ratio were noted in patients with type 2 diabetes (T2D) without DR as compared to the control group [18]. In the studies, more binocular RNFL thickness asymmetry in diabetic patients in relation to the controls without significant differences in RNFL thickness [17,18] was also noted.

In our study, negative correlations in the study group between superior RNFL thickness and duration of diabetes, and between inferior RNFL thickness and mean HbA1c value were observed. Similar to our results, Tekin et al. noticed negative correlations between both global RNFL thickness and macular thickness and HbA1c value and diabetes duration using the SD-OCT study in children with T1D without clinical overt DR. They also found significantly thinner global RNFL thickness as well as temporal and inferior outer macular thickness in relation to the controls [19]. Both overall and superior RNFL thickness correlated with the age of T1D patients in the Dehghani et al. study, at the same time being an indicator of peripheral neuropathy in the patients [20]. In other research, the significant thinning of RNFL and its individual quadrants was confirmed in T2D patients without retinopathy in comparison to the controls [21]. Furthermore, there are also studies in which RNFL thinning was found to correlate with the severity of retinopathy, but it was found also in T2D patients [22]. Interestingly, early structural damage of neuroretina was also related to glucose fluctuations assessed in continuous glucose monitoring (CGM) in the group of T1D patients [23]. In our study, to evaluate glucose variability, the SD for the average blood glucose (BG) level was measured within the two weeks prior to the HD-OCT study. However, it correlated positively only with macular center thickness confirming earlier reports of the effect of glucose fluctuations on the DR presence accompanied by DME [24,25].

In the present study, no differences were found between the groups with respect to other macular full-thickness parameters or GCIPL thickness. Only tendencies to correlation of average GCIPL and lower GCIPL thickness with age and HbA1c were observed. In several studies, however, a reduced GCIPL thickness was found in both pediatric T1D patients and adults with T2D without retinopathy, suggesting that the presence of chronic hyperglycemia may have an early neurodegenerative effect on retinal ganglion cells, even when vascular disorders are not yet present [26,27].

Previous studies on DR pathogenesis have shown that neuronal dysfunction and progressive neurodegeneration are closely correlated with microvascular disorders. According to many authors, the best method to prevent the appearance of diabetic eye disease is correct glycemic control, reducing the risk of developing microvascular complications [28]. However, it seems that local hyperperfusion may be present already at the preclinical stage of DR and may accompany the neurodegeneration progress. In the paper by Golebiewska et al. it was noted that—besides the presence of early neuroretinal lesions—the central choroidal thickness tends to be thicker in T1D children without DR in comparison to healthy children and shows a correlation with gender of the subjects in favor of girls. However, it does not correlate with the duration of diabetes or HbA1c [29]. In our study we did not observe a relationship between CT and the gender, but with the age of the subjects. Moreover, positive correlations between subfoveal and temporal parts of CT with the age of patients were found, both at the time of examination and at the time of diabetes diagnosis.

The Niestrata-Ortiz et al. study carried out in the group of children with type 1 diabetes compared to the age-matched control group also revealed that choroidal thickness is higher in patients with T1D compared to the control group, and increases with the duration of diabetes, which makes it useful for screening in children with type 1 diabetes [30].

Concluding, the results obtained in the group of children and young adults with type 1 diabetes indicate a relationship between age of patients, duration of diabetes, chronic hyperglycemia, glycemia fluctuations and the size of the ONH parameters, thickness of superior and inferior RNFL quadrant, center macular thickness and subfoveal and temporal choroidal thickness.

Our findings can also confirm earlier observations of the researchers that the Cirrus HD-OCT system achieves approximately double the axial resolution of the previous generation Stratus OCT device and allows for better visualization of tissues. This processing HD-OCT software is unique because images are generated by evaluating all of the pixel data to reduce noise and construct the best possible image. This enables visualization and measurement of the full thickness of the choroid and satisfactory repeatable GCIPL thickness measurements using the Cirrus HD-OCT Ganglion Cell Analysis (GCA) algorithm [31,32].

The limitation of our work may be the lack of other—apart from SD for blood glucose level—indicators of glycemia variability, which require CGM system connection to patients, as well as ophthalmic parameters, including the assessment of visual acuity. The obtained results also require further confirmation on a larger group of patients.

However, it seems that simultaneous measurement of choroidal thickness and several neurodegeneration parameters using HD-OCT is useful and the selected parameters may be promising markers of preclinical phase of diabetic retinopathy in patients with type 1 diabetes. This indicates the usefulness of the HD-OCT study as a diagnostic tool and confirms the need for the HD-OCT study in patients with several years of clinical course of type 1 diabetes.

## Figures and Tables

**Figure 1 diagnostics-09-00105-f001:**
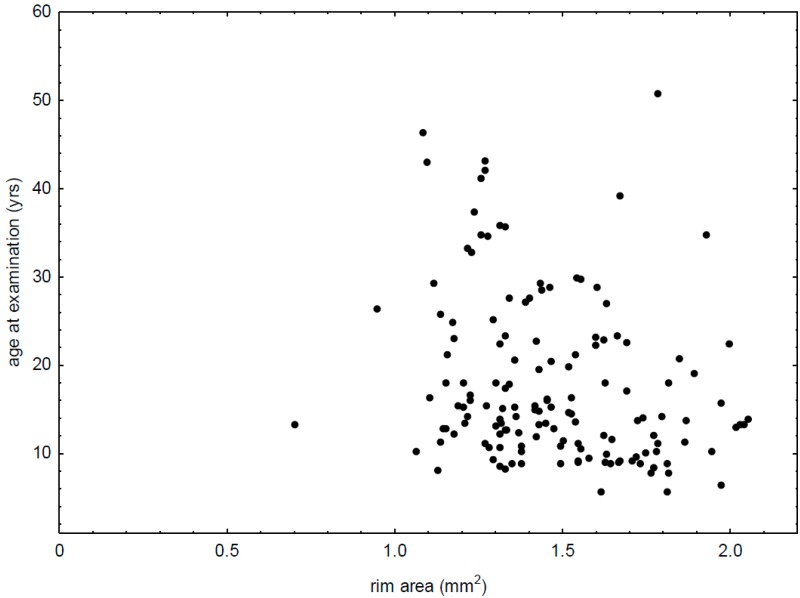
Correlation between rim area and age at examination in patients with type 1 diabetes (R = −0.28, *p* = 0.0007).

**Figure 2 diagnostics-09-00105-f002:**
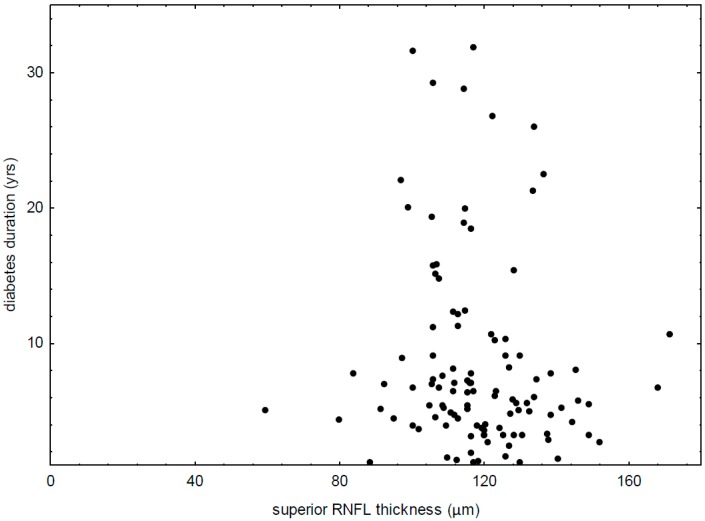
Correlation between thickness of superior retinal nerve fiber layer and duration of type 1 diabetes (R = −0.20, *p* = 0.0336).

**Figure 3 diagnostics-09-00105-f003:**
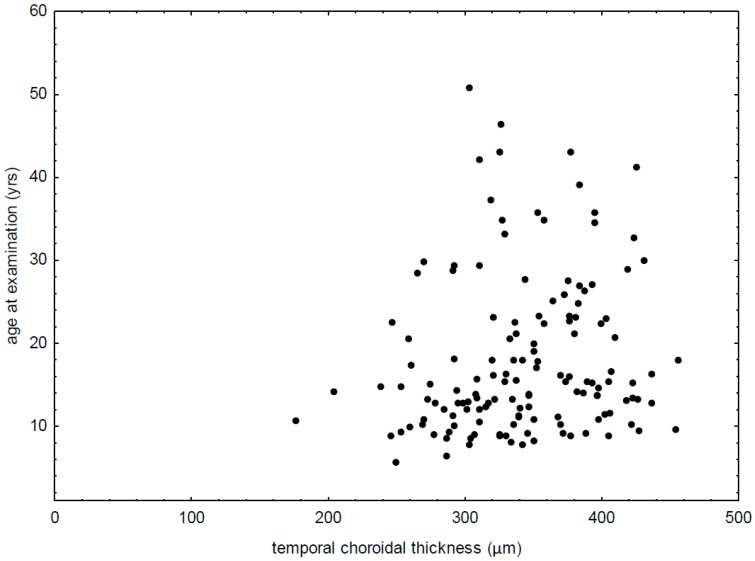
Correlation between temporal choroidal thickness and age at examination in patients with type 1 diabetes (R = 0.21, *p* = 0.0127).

**Table 1 diagnostics-09-00105-t001:** Characteristics of the study group of patients with type 1 diabetes (T1D).

Parameter	Median or Percentage	IQR–Interquartile Range
Age at T1D onset (years)	8.63	4.5–10.8
Diabetes duration (years)	6.02	3.9–10.2
Age at examination (years)	14.47	10.7–20.6
Gender (F/M; %)	55/45	-
Average HbA1c level (%/mmol/mol)	7.10/54.1	6.7/49.77–8/63.9
Average blood glucose level (mg/dL)	155	135–180
Standard deviation for blood glucose level	72	59–79
BMI-Z-score at examination	0.55	−0.36–1.12
Insulin uptake (U/kg)	0.80	0.60–1.07
Total cholesterol (mg/dL)	181.53	153–311
LDL-cholesterol (mg/dL)	110	87–211
HDL-cholesterol (mg/dL)	62	52–98
Triglycerides (mg/dL)	75	55–110
Serum creatinine level (mg/dL)	0.66	0.6–0.7

**Table 2 diagnostics-09-00105-t002:** Comparison of the parameters of ONH, thickness of RNFL and its quadrants, average retinal thickness, central retinal thickness, total retinal volume, CT and GCIPL thickness, assessed using high-definition optical coherence tomography HD-OCT of the study group of patients with T1D and control group.

Parameter	Study Group Median (IQR)	Control Group Median (IQR)	*p* Level
Disc area (mm^2^)	**1.84 (1.62–2.04)**	**1.98 (1.82–2.14)**	**0.0214**
Rim area (mm^2^)	1.43 (1.28–1.63)	1.53 (1.31–1.71)	0.2553
c/d area ratio	0.42 (0.33–0.55)	0.45 (0.32–0.56)	0.6356
Vertical CDR	0.41 (0.32–0.52)	0.43 (0.29–0.53)	0.8136
Cup volume (mm^3^)	0.07 (0.02–0.16)	0.08 (0.03–0.22)	0.4646
RNFL Total Thickness (µm)	93.00 (86.00–101.50)	91.25 (87.50–98.51)	0.4640
Superior RNFL thickness (µm)	116.50 (108.50–128.50)	117.50 (107.00–127.50)	0.6884
Inferior RNFL thickness (µm)	124.50 (109.50–134.50)	122.00 (113.25–128.75)	0.5841
Temporal RNFL thickness (µm)	63.50 (57.50–70.50)	63.75 (55.25–71.00)	0.8605
Nasal RNFL thickness (µm)	69.50 (60.50–76.50)	70.00 (60.5–74.00)	0.6686
Macular average thickness (µm)	283.50 (274.50–293.50)	284.25 (276.25–294.75)	0.4401
Center thickness (µm)	253.25 (241.00–264.00)	255.50 (240.75–266.00)	0.6661
Total macular volume (mm^3^)	10.20 (9.85–10.55)	10.22 (9.95–10.60)	0.3400
Choroidal thickness subfoveal (µm)	352.00 (319.50–393.00)	360.75 (331.50–399.75)	0.3528
Choroidal thickness nasal (µm)	354.00 (318.50–389.00)	359.00 (326.25–376.50)	0.8965
Choroidal thickness temporal (µm)	336.50 (301.50–384.50)	352.75 (314.50–383.00)	0.3070
Average GCLIPL thickness (µm)	83.50 (79.50–88.00)	82.25 (78.75–86.75)	0.6005
Minimum GCLIPL thickness (µm)	81.50 (76.50–86.00)	80.50 (76.75–83.75)	0.7340
GCLIPL superior (µm)	83.50 (79.00–87.50)	82.50 (77.75–87.00)	0.2140
GCLIPL superior-nasal (µm)	85.00 (81.50–89.50)	83.00 (79.50–87.00)	0.1240
GCLIPL inferior-nasal (µm)	84.00 (79.50–88.50)	81.75 (78.00–86.00)	0.1551
GCLIPL inferior (µm)	82.00 (78.00–86.00)	80.75 (77.50–86.00)	0.7551
GCLIPL inferior-temporal (µm)	83.50 (80.00–88.50)	82.75 (78.25–86.25)	0.3227
GCLIPL superior-temporal (µm)	82.50 (78.52–86.50)	81.50 (77.00–84.50)	0.2982

T1D—type 1 diabetes, IQR—interquartile range, ONH—optic nerve head, RNFL—retinal nerve fiber layer, GCILP—ganglion cell layer/inner plexiform layer, CT—choroidal thickness; *p* < 0.05 are indicated in bold.

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
