# Peer review of "The HD-OCT Study May Be Useful in Searching for Markers of Preclinical Stage of Diabetic Retinopathy in Patients with Type 1 Diabetes"

_diagnostics, 2019, doi:10.3390/diagnostics9030105_

Round 1
Reviewer 1 Report
It is an interesting manuscript. Authors succeed to present their data in a clear way adding information to the existing literature. Therefore, I have no corrections to do and the manuscript can be published unaltered.
Author Response
R: Thank you very much for your comment.
Reviewer 2 Report
The authors mention in the introduction about fluorescein angiography being an invasive method to determine DR at clinical setup and they have introduced HD-OCT as an alternative method. As a reader, I was hoping to find out more about the advantages of using this technique, but the authors have not clearly mentioned the advantages of using this method over angiography or SD-OCT. The authors have missed discussing this important aspect of the study which can throw limelight over HD-OCT as a valuable tool. The introduction and the discussion has to be substantially improved.
Author Response
R: Fluorescein angiography (FA) is a test that involves taking a series of photographs of the fundus after intravenous administration of a contrast agent (fluorescein). FA allows visualization of the circulation in the retinal vessels and indirectly the uvea, and allows the assessment of the condition of the retinal pigment epithelium and retinal blood vessels that are not visible during fundus examination. This technique has become fundamental when assessing the complications of diabetic retinopathy, retinal vascular obstruction, or diagnosing many macular diseases. However, FA is associated with severe potential risks. Gastrointestinal disturbances, mainly nausea and vomiting, are the most frequent adverse reactions during FA, which appear in up to 10% of these procedures. Cases of pulmonary edema, myocardial infarction, or anaphylactic shock have also been reported during FA. In addition to these risks, FA may be a time-consuming procedure when peripheral veins are difficult to cannulate, and this enhances patient discomfort.
In an attempt to optimize diagnostic procedures, there is a need to identify the most informative and least invasive diagnostic modality. Optical coherence tomography (OCT) is a safe, non-contact, non-invasive and effective technique of imaging tissue and could be considered as a valid alternative to FA. Intra-retinal leakage is not necessary to confirm the presence of DME, and FA is useful only as a guide for focal or grid laser treatment of thickened areas.
OCT has rapidly emerged as a widely used imaging system in ophthalmology where it is mainly used for diagnosing and monitoring diabetic retinopathy, macular edema, glaucoma and other retinal diseases. Therefore, numerous new diagnostic tools such as high-definition (HD-OCT) and spectral-domain (SD) OCT are being evaluated clinically. These methods provide improved capabilities for the imaging of the intra-retinal morphology and/or allow analysis of the structural and functional aspects of leakage activity. Moreover, they offer realistic 3D imaging of retinal and subretinal levels in various diseases.
Cirrus HD-OCT (Carl Zeiss Meditec, Inc.) is an SD-OCT that has an axial resolution of 5 μm and a scan velocity of 27,000 axial scans per second. These features improve the ability to visualize smaller and thinner structures that are difficult to visualize with time-domain OCT (Stratus OCT). Cirrus HD-OCT has been demonstrated to have high intrasession repeatability in healthy subjects and provides retinal thickness measurements approximately 43 μm higher than does Stratus OCT. The higher axial scan acquisition rate of the Cirrus HD-OCT system allows for reduction of motion artifacts in line scans and rapid acquisition of three-dimensional data volumes. The Cirrus HD-OCT system achieves about double the axial resolution of the previous generation Stratus OCT device and allows for better visualization of tissues. This processing HD-OCT software is unique because images are generated by evaluating all of the pixel data to reduce noise and construct the best possible image. This enables visualization and measurement of the full thickness of the choroid and satisfactory reproducible measurements of GCIPL thickness using the Cirrus HD-OCT GCA algorithm.
The obtained results indicate the usefulness of HD-OCT as a valuable diagnostic tool in the search for early markers of the preclinical phase of diabetic retinopathy and confirm the need for HD-OCT study in patients with several years of clinical course of type 1 diabetes.
We have improved the Introduction and Discussion sections.

Reviewer 3 Report
Summary:
In this paper the authors present their clinical study results which was carried out in order to demonstrate that the diabetic retinopathy in type1 diabetes mellitus is possible to be detected before the clinical manifestation using a non-invasive diagnostic procedure.
The study is original and an interesting, novel and worthwhile work. The authors contribute to this field of research with new, valuable and feasible findings, nevertheless some minor revisions are required before publication.
In the “Material and methods” section, line 68: I think the authors wanted to insert comma instead of dot, like as in line 85. Also in this section, in line 75, instead of “is”, is required “are”. Why is the numbering of the tables reversed? Table 2. In the “Material and methods” and Table 1. in the “Results” section. In the description of study design, the authors stated that the in the study group the patients’ age was between 5.6 and 50.8, nevertheless in Table 2. the “Median or Percentage” of “Age at T1D onset (years)” was calculated to be 8.63. Please, describe the calculation of this finding. Why do the authors use for the verification of the normality of distribution the Kolmogorov-Smirnov test that is not recommended in present-day? In Table 1., “Total macular volume (mm3)” is required to be corrected. Figure 2. and Figure 3. caption font format is required to be uniformed. Please, verify and correct these parts. And finally only as a suggestion: for a better displaying the authors can modify the numbers format in Table 2.
I mean to use a uniform format, for example instead of 253.25 (241 – 264) to write 253.25 (241.00 – 264.00).
Author Response
In the “Material and methods” section, line 68: I think the authors wanted to insert comma instead of dot, like as in line 85. Also in this section, in line 75, instead of “is”, is required “are”.
R: Thank you. We have corrected it.
Why is the numbering of the tables reversed? Table 2. In the “Material and methods” and Table 1. in the “Results” section.
R: This reversal resulted from the fact that in the Instruction for authors it was recommended to present the Results before Material and Methods section. We have changed it.
In the description of study design, the authors stated that the in the study group the patients’ age was between 5.6 and 50.8, nevertheless in Table 2. the “Median or Percentage” of “Age at T1D onset (years)” was calculated to be 8.63. Please, describe the calculation of this finding.
R: This is not a mistake. In the Material and Methods section we have given the minimum and maximum age of patients at the time of examination. All patients in the study group were diagnosed with type 1 diabetes as children, on average about 8 years of age and therefore this is a homogeneous study group.
Why do the authors use for the verification of the normality of distribution the Kolmogorov-Smirnov test that is not recommended in present-day?
R: The assessment of the normality of distribution was carried out using the Kolmogorov-Smirnov test and verified by the Shapiro-Wilk test. We have changed this section.
In Table 1., “Total macular volume (mm3)” is required to be corrected. Figure 2. and Figure 3. caption font format is required to be uniformed. Please, verify and correct these parts. And finally only as a suggestion: for a better displaying the authors can modify the numbers format in Table 2. I mean to use a uniform format, for example instead of 253.25 (241 – 264) to write 253.25 (241.00 – 264.00).
R: Thank you, we have corrected it accordingly.

Round 2
Reviewer 2 Report
Thank you for making the required changes for the manuscript.